# Basivertebral Nerve Ablation for Treatment of Lower Back Pain

**DOI:** 10.3390/biomedicines12092046

**Published:** 2024-09-09

**Authors:** Esther Lee, Joaane Kim, Sadiq Rahman, Neil Daksla, William Caldwell, Sergio Bergese

**Affiliations:** Department of Anesthesiology, Stony Brook University Hospital, Stony Brook, NY 11794, USAjoaane.kim@stonybrookmedicine.edu (J.K.); sadiq.rahman@stonybrookmedicine.edu (S.R.); neilchristopher.daksla@stonybrookmedicine.edu (N.D.); william.caldwell@stonybrookmedicine.edu (W.C.)

**Keywords:** chronic pain, basivertebral ablation, Modic changes, lower back pain, pain management

## Abstract

Lower back pain (LBP) is a widely prevalent global health issue, affecting over half a billion people and remaining the leading cause of years lived with disability (YLDs). LBP significantly impacts healthcare systems, with substantial costs related to surgical procedures and lost workdays. Vertebrogenic back pain (VBP), characterized by specific clinical symptoms and associated with Modic changes (MC) in vertebral endplates, best seen on MRI, is a significant subset of LBP. This paper explores the pathophysiology, diagnosis, and current reports and studies focusing on VBP and the role of basivertebral nerve (BVN) ablation as a therapeutic intervention. Multiple studies, including randomized controlled trials (RCTs) and meta-analyses, demonstrate the efficacy of BVN ablation in reducing pain and improving function in patients with chronic LBP associated with MC.

## 1. Introduction

According to a survey in 2020, low back pain (LBP) affected more than half a billion people worldwide, remaining as the leading cause of years lived with disability (YLDs) globally. While age-standardized rates of LBP have shown a modest decrease since 1990, the absolute number of cases continues to rise, due to population growth and aging, with a projected 843 million prevalent cases (a 36.4% increase) by 2050 [1].

A total of 38.8% (28.7–47.0) of YLDs due to LBP were attributed to occupational factors, smoking, and high BMI. Occupational factors include physically demanding tasks or those involving vibration. Ergonomic factors such as prolonged sitting, standing, bending, or lifting contribute to nearly one-quarter of YLDs due to LBP [1]. Obesity and smoking have demonstrated a positive correlation with the occurrence and persistence of LBP, though a specific causal mechanism has not been elucidated [2].

Due to the high prevalence of LBP, the healthcare utilization and societal burden is significant. LBP comprises a substantial proportion of the caseload for primary-contact disciplines. A review [3] of the epidemiology of LBP in primary care settings states that only about one in three cases of recent-onset LBP episodes resolves completely over a 12-month period. More than half of the recent-onset LBP cases develop into an on-going relapsing pattern of persistent LBP, consuming most of the LBP compensable care resources and showing possible but infrequent or insubstantial positive outcomes [3].

From 2012 to 2014, the direct aggregate costs for all individuals with a spine condition in the USA were USD 315 billion, with a substantial proportion of costs attributed to surgical procedures [1]. In the USA, Australia, and the United Kingdom, the direct costs of back pain represent between 0.19% and 0.42% of GDP, and between 1.65% and 3.22% of all health expenditure [3]. In the USA, 15.4% of the workforce reports an average of 10.5 lost workdays per year due to chronic low back pain, equivalent to approximately 264 million workdays lost [1].

## 2. Vertebrogenic Back Pain (VBP)

Patients with vertebrogenic back pain (VBP) typically complain of lower back pain that is described as a deep, aching, and burning sensation. The pain is usually progressive in nature and may be accompanied by an intermittent electric shock sensation [4]. Many patients experience asymptomatic periods or mild lower back pain interspersed with severe flares that last a few days. Much of this pain is located at the midline L3-S1 levels, with minimal referral cephalically and to the paraspinal and/or gluteal regions [5]. This localization is important in distinguishing vertebrogenic from lumbosacral facet joints and sacroiliac joints as the source of pain, which tend to present with lateralized paraspinal and posterior superior iliac spines region pain, respectively [6]. VBP often worsens with activities involving spinal flexion, such as sitting and bending forward [7]. On physical examination, these patients may have tenderness on percussion and reproduction of pain with spinal-flexion-based movements. While suggestive of VBP, these clinical features are nonspecific and require correlating radiographic findings for an accurate diagnosis.

Basivertebral nerve ablation is targeted to treat lower back pain consistent with Modic changes (MC). Modic changes represent an intraosseous source of pain similar to tumors, stress fractures, etc. This intraosseous source of pain is secondary to degenerative processes of the spine confirmed by end plate changes best detected using MRI, indicating possible edema or inflammation. Vertebral bodies are innervated by intraosseous basivertebral nerves entering through paired neurovascular foramina. Immunostaining of vertebral bodies has shown that they contain substance P, involved in pain pathways. Chronic lower back pain related to these Modic changes is posited to come from vertebral end plate nociceptors activated by basivertebral nerves [8].

The pathophysiology of lower back pain has long been thought to be discogenic in nature. However, growing evidence suggests that the origin of this pain is likely vertebrogenic, which results from damage to the richly innervated end plates [9]. More specifically, damage and degeneration of these endplates allow increased communication between the spinal bone marrow and the disc, causing abnormalities that can be both felt by the patient and visualized on an MRI as Modic changes (MC) [10].

MC are further classified into three types based on their MRI appearance and underlying pathology. MC Type I is characterized by bone marrow edema and active inflammation. It appears as hypointense signal intensities on T1-weighted images (T1W1) and hyperintense signal intensities on T2-weighted images (T2WI). MC Type 2 involves the replacement of bone marrow into yellow fatty tissue. This change is visualized as hyperintense signaling intensities on both T1- and T2-weighted changes. Finally, MC Type 3 is associated with bone sclerosis and appears hypodense on both T1WI and T2WI [11]. Accurate identification of MC on imaging is essential, as only BVN ablation has been shown to be effective in patients with VBP that have the presence of MC on imaging [7].

Clinical symptoms may vary depending on different types of Modic changes (MC). MC are associated with increased pain compared to the controls. This is supported by a study that found a significantly higher number of inflammatory cytokines in patients with MC compared to the controls [12]. Patients with LBP with MC are more likely to experience more frequent and longer LBP episodes and are also more likely to seek care. Lumbar pain is also more frequent and painful in people with MC Type 1 compared to those with Type 2 [13,14].

A newly developed spinal degeneration scoring system recommends evaluating Modic changes, fatty infiltration in the paraspinal muscles, and disc degeneration to predict lower back pain [15]. While MC have been found to be very specific to low back pain, current best practice guidelines require correlating clinical symptoms with radiographic findings indicative of intraosseous changes and damage at the vertebral endplate [16].

## 3. Methods

We conducted a comprehensive review of articles published from 2015 onwards, sourced from PubMed Central. Our focus was on studies that investigated the effectiveness of basivertebral nerve ablation in managing chronic low back pain for at least six months. This included randomized controlled trials, clinical trials, and meta-analyses. To be eligible for consideration, studies had to meet several criteria. Participants needed to have chronic low back pain lasting at least six months and must have been unresponsive to conservative treatment options. Additionally, studies were required to provide imaging evidence of basivertebral back pain with Type 1 or Type 2 Modic changes, as identified through magnetic resonance imaging (MRI). The primary scales of measurement used in the studies were the Oswestry Disability Index (ODI) and the Visual Analog Scale (VAS). The main characteristics of each study are summarized in Table 1.

## 4. Current Reports and Studies

Becker et al. [17] conducted the earliest known BVN ablation clinical study to demonstrate the efficacy of BVN ablation in the treatment of chronic lumbar back pain. Patients with more than 6 months of low back pain with Modic Type I or II changes who were unresponsive to at least 3 months of conservative care underwent radiofrequency energy ablation. Patient-reported outcomes were gathered using the Oswestry Disability Index (ODI) questionnaire to assess functional disability and the Visual Analog Scale (VAS) to measure a patient’s pain intensity. Follow-up at 3 months showed a mean ODI decrease from a baseline of 52 ± 13 to 23 ± 21 (*p* < 0.001). The significant improvement in ODI persisted throughout the 1-year study period. In addition, the mean baseline VAS score decreased from 61 ± 22 to 45 ± 35 at the 3-month follow-up (*p* < 0.05). These findings not only confirmed the involvement of the basivertebral nerve in causing axial back pain, but also suggested that targeting this nerve could yield lasting pain relief. The study also demonstrated possible adverse events that may be associated with BVN ablation therapy. In the immediate postoperative period, four patients experienced non-serious, device- or procedure-related adverse events in the form of buttock pain, lumbar pain, dysesthesia, and mild transient thigh numbness, requiring pain medication. Importantly, none of the 16 patients reported significant functional neurologic deficits or serious adverse events related to the treatment. Despite these promising findings, one of the weaknesses of this study was the lack of a viable control group. Nevertheless, this was a promising start of utilizing BVN ablation as a treatment for chronic low back pain, highlighting the need for further research in this area [17].

The SMART trial conducted by Fischgrund et al. [18]. is the largest known BVN ablation clinical study. The study was a randomized double-blinded multicenter study of 225 patients with chronic lumbar pain that were randomized into BVN ablation or sham treatment. As patient’s response to treatment is believed to be influenced by both central brain responses through the modification of expectations and peripheral responses, this study utilized the sham group not only to assess the treatment group’s effectiveness, but also to investigate the role of central brain response as part of the pain response. The patient population was required to have chronic lower back pain (CLBP) for over 6 months that was unresponsive to non-operative management, a minimum ODI of 30 and VAS of 4 mm, and Type 1 or 2 Modic changes on imaging. The sample had a mean baseline ODI of 42 and were randomized 2:1 to BVN thermal ablation (n = 147) or sham treatment (n = 78). After 3 months, there was a significant difference in the decrease in the average ODI between the groups, with the treatment arm being 20.5 and the placebo being 15.2 (*p* = 0.019). There was a significant sustained difference between the two groups for VAS as well. These findings suggest that, although the pain response results from both the intervention and placebo effects, BVN ablation, which targets peripheral responses to pain, holds significant therapeutic value [8]. Importantly, MRI evaluations at the six-week and six-month follow-up periods did not reveal any evidence of spinal cord abnormalities, avascular necrosis, or accelerated disc degeneration. One patient did experience a change from Modic Type 1 to 2 between the six-week and six-month period.

Fischgrund et al. conducted a further study measuring the clinical outcomes of the previously studied sample over a 2-year period [19]. Those who had undergone BVN were followed for safety at 18 months and 24 months. Of note, all participants continued conservative therapy as prescribed at the baseline through the study duration, although specific methods of conservative treatment were not specified. Of the 78 patients in the sham arm, 57 (73%) chose to receive BVN and were followed for 12 months, while those who did not cross over (n = 18) were also followed for a total of 24 months. The mean improvements in ODI and VAS compared to the baseline at 2 years were 23.4 and 3.57, respectively. The responder rates, defined as ODI ≥ 10-point improvement and VAS ≥ 1.5 cm improvement, were 76.4% and 70.2%, respectively. Moreover, 57.5% reported ODI ≥ 20-point improvement. The study demonstrated sustained clinical benefits in ODI and VAS over the two-year follow-up. Among 27 participants who were taking opioids to manage CLBP at enrollment, 60.7% reduced their opioid medication, and 46.4% eliminated opioid use during the study period. In addition, only 8 (7.5%) received spinal injections for either new onset or persistent LBP after the treatment, a significant decrease from the 61 (57.5%) who had reported prior injections at the baseline. In the first year of follow-up, two patients treated with radiofrequency (RF) ablation required surgical spine intervention for persistent lower back pain and new disc herniation at the L2–L3 level, outside of the treated level in the study. In addition, during the second year of follow-up, eight RF-ablation-treated patients opted to leave the study to undergo surgical therapies, with seven receiving lumbar fusion procedures at the same level as the study procedural site. These findings suggest that patients can have residual pain and may require further surgical intervention even after BVN ablation therapy. The INTRACEPT study was a parallel, open-label RCT [20] conducted at 20 US sites comparing BVN to standard care (SC) for the treatment of patients suspected to have vertebrogenic CLBP. A total of 140 patients with chronic LBP with Modic Type 1 or 2 changes were studied. The mean baseline ODI was 46.1 and VAS was 6.67, with 67.3% of them having experienced LBP symptoms for more than 5 years. While the sample population was similar to that of Fischgrund et al., the inclusion criteria were revised to involve patients with a history of laminectomy or discectomy and those with moderate symptomatic spinal stenosis. The patients were randomized 1:1 to either a radiofrequency ablation of the BVN therapy or continued SC, which consisted of pain medications, physical therapy, exercise, chiropractic treatment, acupuncture, and spinal injections. The mean changes in ODI comparing the RF ablation arm to the standard arm at 3 months were −25.3 points vs. −4.4 points, resulting in an adjusted difference of 20.9 points (*p* < 0.001). The mean changes in VAS were −3.46 vs. −1.02, with an adjusted difference of 2.44 cm (*p* < 0.001). A total of 74.5% of patients in the RF ablation arm achieved a ≥10-point improvement in ODI, compared with 32.7% in the standard care arm (*p* < 0.001). The significant difference at the three-month endpoint led to the halt of further patient enrollment and the initiation of early cross-over from the control arm to BVN ablation. Among 51 patients who were subjected to BVN ablation therapy, 39 (78%) noted satisfactory improvement in their pain, while 8 (16%) reported no change and 3 (6%) experienced a worsening of pain.

As an extension of the 3-month interim analysis reported in the INTRACEPT study, Smuck et al. [21] presented the results of the RCT of BVN ablation versus SC at 3 and 6 months and results of the treatment arm at 12 months. SC patients who elected to receive BVN ablation after halting SC were re-baselined and followed up for 6 months post BVN ablation. Consistent with the interim analysis, this study demonstrated a change of −20.3 in ODI from SC (*p* < 0.001) and −2.5 in VAS pain scale (*p* < 0.001) at the 3-month follow-up. At 12 months, statistically significant and clinically meaningful improvements in pain and function in the BVN ablation compared to the baseline were maintained for all patient-reported outcomes, including mean ODI, VAS, and quality of life measurements EQ-5D-5L SF-36 and patient satisfaction questionnaire. A total of 44.3% of patients reportedly achieved a >75% reduction, and 29.5% reported 100% pain relief at 12 months. Notably, former SC arm patients that crossed over to BVN ablation also reported statistically significant (*p* < 0.001) and clinically meaningful improvements in pain and function at 3 and 6 months after BVN ablation from re-baseline. There was a 24.7 ± 18.5-point change in mean ODI at 3 months and a −25.9 ± 17.1-point change at 6 months. The mean VAS reduction was −3.5 ± 2.6 and −3.8 ± 2.9 at 3 and 6 months, respectively. In addition, the patient satisfaction questionnaire showed that 74% of patients treated with BVN ablation had an improvement in pain, with 29.5% of them reporting complete resolution of pain at 12 months. Notably, among 54.5% of the BVN ablation arm that received an epidural steroid injection within the 12 months prior to involvement, only 4.5% (3/66) received a steroid injection in the same region, compared with 18% (11/61) in the SC arm in the 12 months after ablation. None of the patients who crossed over to receive BVN ablation from the former SC arm required epidural steroid injection in the six-month follow-up, highlighting adequate control of CLBP after the procedure. The INTRACEPT study demonstrates the superiority of BVN ablation over standard of care in improving patient-reported outcomes up to 12 months, which was demonstrated with the original treatment arm and the former SC arms who chose to undergo BVN ablation [21].

In a prospective, single-arm, multicenter, open-label case series, 28 patients with chronic low back pain (CLBP) of at least six months duration and with MC1 or MC2 vertebral endplate changes were treated with radiofrequency ablation of the BVN in up to four vertebral bodies. Notably, the patients in this study were treated in a typical community spine clinic setting by providers with no special expertise in this procedure with the goal of creating a “real-word” patient population. In addition, patient selection was looser compared to prior studies. Other than the Modic changes shown on MRI and clinician judgement, there were no requirements to rule out other sources of the pain. The maximum number of VB levels treatable was increased from three to four, and the treatment did not have to involve consecutive levels. No upper limits on ODI or VAS were imposed. More than half of the patients previously underwent at least one trial of physical therapy, chiropractic care, or spinal injections, and 14.3% had undergone prior discectomy. The clinical outcomes were consistent with those reported in the treatment arm of the efficacy RCT. For example, 3-month post-treatment outcomes, as measured by ODI and VAS, showed a mean reduction of −30.07 ± 14.52 (*p* < 0.0001) and −3.50 ± 2.33 (*p* < 0.0001), respectively. Additionally, 21 (75%) reported an improvement in their condition, 5 (18%) reported no change, and 2 (7%) complained of worsening pain. The study also reported that, of the 86% of patients who were working either full-time or part-time, 83% reported improvements in work function from the baseline. Despite involving community clinics with non-experts performing the procedure and broadening the patient selection criteria, there were significant improvements in both pain and work functionality, suggesting that the study findings may be generalizable to the broader CLBP population [22].

Continuing the exploration of the effectiveness of BVN ablation in a broader patient population, a prospective, single-arm, open-label effectiveness trial of 48 patients from community spine and pain practices was studied. The inclusion criteria required more than 6 months of CLBP and Type 1 and 2 Modic changes on MRI, but no limits on the baseline patient-reported outcomes (ODI and VAS) were enforced. Patients with prior discectomies and extended-release opioid use were allowed to participate, and there were no requirements for conservative treatments that must have been attempted prior to enrolling. The mean ODI reduction in the 45 patients that completed 12 months of follow-up was 32.31 ± 14.07 (*p* < 0.001), with 88.89% (40/45) patients reporting ≥15-point ODI decrease. The mean VAS pain score decrease was 4.31 ± 2.51 at 12 months (*p* < 0.001). Similarly, SF-36 and EQ-5D-5L scores improved, being 26.27 ± 17.19 and 0.22 ± 0.15 (each *p* < 0.001), respectivley. There was a remarkably high satisfaction rate from patients, as follows: 84.4% rated their condition as improved, while 11% reported no change, and 4% indicated a worsening of pain. Additionally, among 85% of participants who were employed full-time, 21% of them (10/47) reported missing work due to lower back pain. A significant reduction of more than 70% was seen, with only 6.7% (3/45) missing work due to lower back pain at the 12-month follow-up. There was also a significant reduction in post-procedural opioid requirement, as, of the 10 (21%) of patients taking opioids at the baseline, only 3 (6.7%) reported actively taking opioids at the 12-month follow-up. Similarly, only one patient required epidural injections in the 12 months after the ablation therapy, compared to 49% (23/47) prior to the baseline. Thus, the study concluded that BVN ablation was effective in community practices and maintained its efficacy in broader clinical application for 12 months [23]. Conger et al. [24] conducted a meta-analysis of 12 single-arm studies looking at patients with CLBP associated with Modic 1 and 2 changes on MRI that underwent intraosseous BVN ablation. These studies had two randomized controlled trials and four single-group cohort studies for 414 participants. Most study participants were Caucasian, nonobese, college educated, and employed and were in their mid-40s to early 50s with ≥5 years of LBP (62–72%). The reported measured outcomes ranged between 3 and 60 months. The primary outcome was the proportion of individuals with ≥50% pain improvement on VAS/NRS at 6 and 12 months, and the secondary outcome consisted of the composite responder rates defined as ≥15-point reduction in ODI and ≥2 cm reduction in VAS. Significant differences were observed at 3 and 6 months between the arms, being 65% (95% CI 51–78%) and 64% (95% CI 43–82%), respectively. The secondary outcomes measuring the composite responder rate were 75% (95% CI 63–86%) and 75% (95% CI 63–85%) at 6 and 12 months, respectively. The study concluded that there is moderate-quality evidence that intraosseous BVN RFA effectively reduces LBP and related disabilities compared with sham RFA and continued standard care treatment according to the GRADE criteria [24].

## 5. Discussion

The cited studies include a chronological report of the significant trials assessing the clinical efficacy and safety of BVN ablation. The results of these studies support BVN ablation over the standard care of CLBP. All demonstrated statistically significant improvement of the mean ODI and VAS in the treatment arm compared to the control group. Moreover, BVN ablation groups showed a significant increase in responder rate defined as ≥10- or 15-point ODI and ≥2-point VAS, supporting clinical improvement [17,18,19,20,21,22,23].

The quality of life of post BVN ablation was also measured using EQ-5D-5L and SF-36 scales, which demonstrated significant improvement in favor of the treatment group [18,19,20,21,22,23,24]. The SF-36 scale can be further broken down into the physical and mental component scores and demonstrated improvements in both categories [20,21,22]. Additional questionnaires were given asking about the satisfaction of the procedure. In Smuck et al.’s study [21], 74% of BVN ablation arm patients reported improvement in their conditions and considered the treatment a success. Similar questions were asked by Khalil et al. [19], with 78% of the treatment arm stating improvement and 88% stating that they would recommend the procedure to others with the same condition [19].

Although these studies reported sustained improvement from the baseline up to 12 months, longer follow-up is warranted in order to understand the long-term effects of BVN ablation. Fischgrund et al. [18] continued to follow the previously studied sample over a 2-year period and observed that the rates for ODI and VAS were maintained with a responder rate of 70.2% of patients. Furthermore, Fischgrund et al. [18]. released an updated report the following year, looking at 100 patients who had undergone treatment with a mean follow-up of 6.4 years (5.4–7.8 years) and saw that the composite responder rate using thresholds of ≥15-point ODI and ≥2-point VAS at 5 years was 75%. While there are sparse studies that follow patients beyond 12 months, Fischgrund et al. support the sustained effects of BVN ablation even after 5 years.

Patient reports of work functionality and satisfaction also provide valuable insights into assessing the efficacy of BVN ablation. Looking at the impact of BVN ablation on work function, Trummees et al. [22] noted that 86% of subjects were working at the baseline. The number of patients who missed work due to CLBP decreased from six cases to one at 3 months post-procedure, while those who reportedly spent >1/2 day in bed due to LBP in the two weeks prior to the baseline decreased from seven cases to two. Similarly, Macadaeg et al. [23] saw that, of the 85% of subjects (n = 47) who were working full-time, patients missing work for LBP decreased by 70%, with a reduction in days missed from an average of 2.5 to 2 days. In addition, an initial survey showed that 15 patients averaged 1.04 days in the past two weeks where they spent more than half of the day in bed. This was reduced by 87% to two patients at 12 months [23]. In four studies, a significant percentage of participants reported satisfactory improvements in their condition after the BVN ablation therapy [20,21,22,23]. In comparison, 6% (3/51), 7% (2/28), and 4% (2/47) of treatment group participants, a relatively small portion, reported a worsening of pain after the procedure [19,22,23]. These studies suggest that, while BVN ablation provides effective pain relief and thereby improves functionality overall, a small portion of patients may experience post-procedural residual pain. Although the specific characteristics of reported post-procedural residual pain have yet to be described in the literature, this highlights the potential need to utilize multiple therapies to effectively address chronic low back pain in conjunction with BVN ablation. The impact of BVN ablation on opioid utilization is unclear. According to the meta-analysis by Conger et al. [24], opioid use decreased in participants over time after BVN RFA in most studies but did not differ significantly from the sham or standard-of-care groups at 3–12 months. In the SMART trial, active opioid utilization decreased from 30/100 to 4/100 by the end of 5 years, but no comparison to a placebo group was given [25]. Similarly, Macadaeg et al. noted a decrease in opioid use from 10 patients (21%) at the baseline to only 3 (6.7%) at the 12-month follow-up, but there was no control group for comparison as it was a single-arm study. Overall, healthcare utilization appears to decrease substantially after BVN RFA, with a low rate of further intervention or surgical treatment.

Three studies noted a decrease in epidural steroid injection use in the affected region post BVN ablation observed up to two years [21,22,23,25]. Post-procedural intervention and surgical treatment were seldom in the published studies either. In the SMART trial, only 8% of the total number of patients (n = 100) progressed to fusion and 3% had a facet RF ablation performed during the five years of follow-up. It is noteworthy that five of the eight patients that progressed to fusion were at a single study site, which may not be reflective of future outcomes in most clinical practices [25]. One patient [23] underwent facet rhizotomy at the same level as treatment seven months post BVN ablation.

The rate of adverse events in the BVN ablation arm remained low among the reported studies. There were no cases of delayed complications or death from the procedure. Compared to other studies, Becker et al. reported the highest rate of adverse events (4 out of 16 cases), which consisted of buttock pain, lumbar pain, dysesthesia, and mild transient thigh numbness that were resolved with pain medication and had no significant functional or sustained neurologic deficits. As this was a pilot study, these findings could indicate that the INTRACEPT procedure was in the earlier stage of technique refinement, with overemphasis of adverse events due to the small sample size. Indeed, with larger RCT trials, the complication rate was 2.7% (8/225 cases) and 10.2% (13/61 cases) [18,21]. For the middle-sized studies, the complication rate ranged between 13.7% (7/51 cases) and 17.9% (5/28 cases), which consisted of radiculitis and aborted surgeries due to an inability to access the pedicle [22,23]. The most commonly reported adverse event is transient leg pain, which is thought to be secondary to pedicle breach. All cases were resolved, with most successfully treated with oral steroids [24]. According to the INTRACEPT study [20], the median time to resolution for the leg pain was 48.5 days. Two cases of retroperitoneal hemorrhage have been reported in the SMART trial, which may be due to excessive lateral positioning resulting in the violation of the lumbar segmental artery [22]. Other adverse events included vertebral compression fracture (n = 1), incisional pain, urinary retention, and dysesthesia [24]. Overall, there were no pathological structural changes in the vertebral bodies, intervertebral discs, or joints associated with BVN ablation therapy in the long term.

BVN ablation is not the only percutaneous interventional therapy available for CLBP. A meta-analysis was conducted comparing the efficacy of various treatments, including radiofrequency ablation of the basivertebral nerve, annuloplasty, steroid injections into the disk, facet joint and medial branch, biological therapies, and multifidus muscle stimulation. This comparison was based on changes in VAS and ODI scores in patients with low back pain lasting over six months due to degenerative spinal conditions, with a follow-up period of up to 24 months. BVN ablation was selected as the basis for comparison due to its significant improvements in VAS and ODI scores across all follow-up periods. Only biological therapy and multifidus muscle stimulation are performed similarly to BVN ablation [26,27]. Although there were no statistically significant differences in the VAS scores at the 6- and 12-month follow-ups for radiofrequency ablation and facet steroid injections, the improvements in the ODI scores were significantly less. Interestingly, the study highlighted that BVN ablation, biological therapy, and multifidus muscle stimulation each targeted different pathologies of CLBP, suggesting the potential for a treatment algorithm tailored to the source of the back pain [28].

One should acknowledge the strict patient selection criteria that the landmark trials Khalil and Fischgrund used for their sample population [18,19]. Truumeees and Macadaeg conducted their studies in a community spine clinic with looser criteria and demonstrated a similar success rate [22,23]. This supports the potential application of BVN in the general CLBP population; however, additional studies should be conducted. However, through these studies, a more detailed description of the characteristic pain patterns associated with vertebrogenic LBP has been developed by analyzing the outcomes of patients who achieved pain relief or functional improvement following BVN RFA. This enhanced understanding can aid in the clinical assessment of procedure candidacy.

It is important to note that these reported studies were all industry-funded [24]. The studies support each other in the similarity of results and replicability of design, but the source of funding calls into question publication bias. There are non-industry-funded studies in the literature, however, they consist of small sample sizes. With the rising popularity of BVN ablation as CLBP treatment, non-industry-funded, multicenter studies consisting of a large patient population should be considered.

## 6. Conclusions

Significant advancements have been made in understanding and treating chronic lower back pain, particularly through the vertebrogenic model, which attributes such pain to the basivertebral nerve. Thus, radiofrequency ablation of the BVN has emerged as a promising treatment for patients with vertebrogenic lower back pain. The efficacy, safety and even superiority of this treatment compared to SC have been well documented in the literature. However, many of these studies have been performed on patient populations selected using narrow inclusion criteria. Future studies should aim to address this limitation by determining whether BVN ablation results in the same clinical outcome in a broader patient population can be found. Furthermore, with the increasing need to reduce opioid use, additional efforts must be made to understand the association between opioid use and BVN ablation outcomes.

To date, the INTRACEPT procedure is the only Food and Drug Administration (FDA)-cleared platform for BVN ablation, specifically targeting the L3 through S1 vertebrae. This procedure is indicated for patients with axial lower back pain that persists over six months who are refractory to conservative treatment and show MC Type 1 and/or Type 2 on imaging. Given the strict criteria for this FDA-approved procedure, more research is necessary to demonstrate its efficacy and safety in a broader patient population. Currently, there are no FDA-approved alternative techniques to the BVN ablation procedure other than the transpedicular approach using a radiofrequency probe. This method has sometimes been shown to be associated with side effects such as leg pain from pedicle breech. Future research should focus on developing safer techniques and tools, such as smaller probes, to minimize the risk of complications [7].

A growing amount of evidence suggests that radiologic modalities, aside from MRI, may be helpful for identifying vertebrogenic pain. For example, MR spectroscopy and novel MRI sequences, such as IDEAL and UTE, have shown potential in isolating the source of back pain and differentiating between annular pain and vertebrogenic pain [29]. Additionally, single-photon emission-computed tomography (SPECT) is a hybrid radiographic technique by which a bone scan with radiotracer uptake is overlaid on three-dimensional CT imaging. Inflamed and metabolically active endplates, characteristic of vertebrogenic pain, show increased radiotracer uptake on SPECT. Some studies have already demonstrated the significantly increased radiotracer uptake on SPECT in patients with Type 1 Modic change. Given the utility of CT imaging for correctly identifying pathological changes underlying vertebrogenic pain, SPECT may be further explored as an alternative diagnostic tool when MRI is contraindicated or not feasible [7].

There is ongoing research in serum biomarkers linked to vertebrogenic pain that may be promising. Additionally, objective monitoring of real-life functionality and physical performance using wearables seems to be an important area for future research, as they can help to identify kinematic and behavioral markers of spine disease. Understanding low back pain as a multidimensional pathology requires investigation through various lenses in order to better comprehend the diagnosis and influence treatment paradigms. Comprehensive approaches that integrate patient history, clinical presentation, advanced imaging, and newly developed techniques will likely enhance the diagnosis and management of vertebrogenic pain.

## Figures and Tables

**Table 1 biomedicines-12-02046-t001:** Study Characteristics.

Authors (Year)	Journal	Groups Studied and Intervention	Study Design & Method	Inclusion Criteria	Exclusion Criteria	Results and Findings	Adverse Events
Becker et al. 2017 [17]	The Spine Journal	16 patients with chronic (more than 6 months), lower back pain unresponsive to at least 3 months of conservative care were treated after confirmatory MRI and discography.	Consented and enrolled patients underwent radiofrequency energy BVN ablation guided in either a transpendicular or extrapedicular approach performed at 3 US sites. Preoperative screening used MRI finding of Modic type I or II changes and positive confirmatory discography to determine the affected levels. Follow up at 6 weeks, and 3, 6, and 12 months postoperatively.	CLBP greater than 6 months that is unresponsive to at least 3 months of conservative care. Type 1 or Type 2 Modic changes limited to the L3, L4, L5, and S1 vertebrae.	Prior spinal surgery, spondylolisthesis, scoliosis, history of spinal infection, prior spinal malignancy, patients who had radicular symptoms or identified pathology at more cephalad levels.	Mean baseline ODI decreased from 52 ± 13 from 23 ± 21 at 3 months (*p* < 0.001). Mean baseline VAS decreased from 61 ± 22 to 45 ± 35 at 3 months follow-up (*p* < 0.05), and the mean baseline physical component summary increased from 34.5 ± 6.5 to 41.7 ± 12.4 at 3 months follow-up (*p* = 0.03).	In the immediate post-operative period, 4 patients experienced non-serious, device- or procedure-related adverse events in the form of buttock pain, lumbar pain, dysesthesia, and mild transient thigh numbness, only requiring pain medication. No significant functional neurologic deficits or serious adverse events related to treatment were reported.
Fischgrund et al. 2018 [18]	Eur Spine J	225 patients diagnosed with chronic lumbar back pain were randomized to BVN ablation (n = 147) or sham treatment (n = 78).	Patients were randomized 1:1 to receive RF ablation or to continue standard care in 15 outpatient sites in the US and 3 in Europe. Self-reported patient outcomes were collected using validated questionnaires. RF patients were followed at 6 weeks, and 3, 6, 9, 12, and 24 months. Standard care patients were followed at 3, 6, 9, and 12 months.	Skeletally mature patients with chronic (greater than 6 months), isolated lumbar pain, who had not responded to at least 6 months of non-operative management Type 1 or Type 2 Modic changes at a minimum of two and maximum of three consecutive vertebral body from L3-S1. Candidates had to report minimum ODI of 30 points and minimum VAS of 4 cm.	Radicular pain, previous lumbar spine surgery, symptomatic spinal stenosis, diagnosed osteoporosis (T < 2.5), disc extrusion or protrusion > 5 mm, spondylolisthesis > 2 mm at any level, 3 or more Waddell’s signs of Inorganic Behavior, and a Beck Depression Inventory of greater than 24.	The average decrease in ODI was 20.5 and 15.2 points in the treatment and sham groups, respectively (*p* = 0.019). 75.6% of patients in the treatment group exhibited clinical improvement at 3 months.	1 device-related adverse event occurred in a sham patient, who developed vertebral compression fracture in the setting of osteopenia. 8 procedure-related events were reported in a total of 225 index procedures for a complication rate of 2.7%. The events included nerve root injury (n = 1), lumbar radiculopathy (n = 2), retroperitoneal hemorrhage (n = 1), and transient motor or sensory deficits (n = 4).
Fischgrund et al. 2019 [19]	Int J Spinal Surg	225 patients diagnosed with chronic lumbar back pain were randomized to BVN ablation (n = 147) or sham treatment (n = 78).	Patients were randomized using 2:1 block randomization to either treatment or sham arm in 15 outpatient sites in the US and 3 in Europe. Patients were evaluated preoperatively, and at 2 weeks, 6 weeks, and 3, 6, 12, 18, and 24 months postoperatively. Patients randomized to the sham control arm were allowed to cross to the treatment group at 12 months. Due to a high rate of crossover, RF ablation treated participants acted as their own control in a comparison to baseline for the 24 month outcomes.	Same as Fischgrund et al. 2018 [18]	Same as Fischgrund et al. 2018 [18]	Patients in the RF group showed statistically significant improvement in the ODI, visual analog scale (VAS), and Medical Outcomes Trust Short-Form Health Survey Physical Component Summary at all follow-up points through 2 years. Compared to baseline, the mean percentage improvement in VAS and ODI at 2 years was 52.9 and 53.7%, respectively	There were no device or procedure-related patient deaths, no unanticipated adverse device effects, and no device-related serious adverse events (SAEs) reported in the study. 1 device- related AE occurred in a sham patient, who crossed over to the active treatment at 1 year. The patient was being treated with high levels of hormone replacement therapy and developed a vertebral compression fracture. However, further diagnostic workup revealed concomitant osteopenia.
Khalil et al. 2019 [20]	Spine J	140 patients with chronic lumbar back pain with vertebral endplate changes between L3 and S1 were randomized to undergo standard care (n = 53) or radiofrequency (RF) ablation of the BVN (n = 51).	Patients were randomized 1:1 to receive RF ablation or to continue standard care in 20 US sites. The primary endpoint of the study was collected at 3 months post randomization (standard care) or post-treatment (RF ablation). However, all RF ablation patient were followed at 6 weeks, and 3, 6, 9, 12, and 24 months. Standard care patients were followed at 3, 6, 9, and 12 months.	Skeletally mature patients with chronic (greater than 6 months), isolated lumbar pain, who had not responded to at least 6 months of non-operative management Type 1 or Type 2 Modic changes at a minimum of two and maximum of three consecutive vertebral body from L3-S1. Candidates have to report minimum ODI of 30 points and minimum VAS of 4 cm.	MRI evidence of Modic at levels other than L3–S1. Patients with radicular pain, previous lumbar spine surgery (discectomy/laminectomy allowed if >6 months before baseline and radicular pain resolved), symptomatic spinal stenosis, metabolic bone disease, spine fragility fracture history, or trauma/compression fracture, or spinal cancer, spine infection, active systemic infection, bleeding diathesis were excluded. Patients with radiographic evidence of other pain etiology, such as disc extrusion or protrusion > 5 mm, or spondylolisthesis > 2 mm at any level spondylolysis at any level were also excluded. Other exclusion criteria for patient selection included: facet arthrosis/effusion correlated with facet-mediated LBP, Beck Depression Inventory > 24 or 3 or >Waddell’s signs, compensated injury or litigation, those currently taking extended release narcotics with addiction behaviors, BMI > 40, Bedbound or neurological condition that prevents early mobility or any medical condition that impairs follow up, or contraindication to MRI, allergies to components of the device, or active implantable devices, pregnant or lactating.	Patient-reported outcome measures were significantly higher in the RF group; 74.5% of patients in the RF group had ≥10-point improvement in the Oswestry Disability Index (ODI), compared with 32.7% in the standard care group. Changes in ODI scores for the RF and control groups at 3 months were −25.3 and −4.4 points, respectively.	3 patients reported general procedure-related events such as incisional pain, urinary retention, and lateral femoral cutaneous neurapraxia. 7 reported procedure related adverse events: one patient had back pain in a new location, while the remaining 6 had either leg pain or paresthesia. All adverse events were categorized as mild, and did not require interventions beyond oral pain medications.
Smuck et al. 2021 [21]	Reg Anesth Pain Med	140 patients with CLBP with vertebral endplate changes between L3 and S1 were randomized into BVN ablation (n = 66) vs. SC (n = 74).	Patients were randomized 1:1 to receive RF ablation or to continue standard care in 23 US sites. Self-reported patient outcomes were collected using validated questionnaires. RF patients were followed at 6 weeks, and 3, 6, 9, 12, and 24 months. Standard care patients were followed at 3, 6, 9, and 12 months.	Same as Khalil et al. 2019 [20]	Same as Khalil et al. 2019 [20]	At 3 months, mean ODI reduction showed a difference between arms of −20.3 (*p* < 0.001) and VAS difference of −2.5 cm between arms (*p* < 0.001). At 12 months, basivertebral ablation showed a 25.7 ± 18.5 point reduction in mean ODI (*p* < 0.001), and a 3.8 ± 2.7 cm VAS reduction (*p* < 0.001) from baseline. Former SC patients who elected BVN ablation (n = 61) demonstrated a 25.9 ± 15.5 point mean ODI reduction (*p* < 0.001) from baseline.	5 patients complained of postoperative pain related to positioning of procedure & incisional pain. 13 (10.2%) non-serious device-procedure-related leg pain events (66 treatment arm and 61 BVN ablation). One urinary retention, one nausea, one skin rash from prep solution, one corneal abrasion related to surgical eye protection and one incision infection.
Truumees et al. 2019 [22]	Eur Spine J	28 patients with CLBP with vertebral endplate changes between L3 and S2 received BVN ablation	BVN performed in 2 community spine and pain practices. Self-reported patient outcomes were collected using validated questionnaires. RF patients were followed at 6 weeks, and 3, 6, 9, 12, and 24 months. Standard care patients were followed at 3, 6, 9, and 12 months.	CLBP duration of greater than 6 months with conservative treatment and Modic type 1 or type 2 changes from L3 to S2	Symptomatic spinal stenosis, disk protrusion > 5 mm, spondylolisthesis > 2 mm at any level, radiculopathy, previous lumbar spine surgery (except discectomy/laminectomy if >6 months prior to baseline), spinal cancer, spine infection, radiographic evidence of other pain etiology, BMI > 40	Mean change in ODI at 3 months posttreatment was −30.07 +14.52 points (*p* < 0.0001); mean change in VAS was −3.50 + 2.33 (*p* < 0.0001) that remained statistically significant at 6 months.	5 AEs reported; none were serious or device-related. One was an aborted procedure due to inability to access. Two were mild cases of leg pain with potential pedicle tract issues.
Macadaeg et al. 2020 [23]	N Am Spine Soc J	48 patients with CLBP with vertebral endplate changes between L3 and S1 received BVN ablation	Open-label, single-arm trial with patients treated with intraosseous RF ablation of the BVN using Intracept system at each level that exhibited qualifying Modic changes with continuation of nonsurgical therapy via clinician’s judgement performed in 2 community spine and pain practices. Patients followed up for a minimum of 12 months.	CLBP with a duration of greater than 6 months with non-surgical management and Modic Type 1 or 2 changes from L3 to S1	Symptomatic spinal stenosis, disk protrusion > 5 mm, spondylolisthesis > 2 mm at any level, radiculopathy, previous lumbar spine surgery (except discectomy/laminectomy if >6 months prior to baseline), spinal cancer, spine infection, radiographic evidence of other pain etiology, BMI > 40	Mean reduction in ODI at 12 months was 32.31 ± 14.07 (*p* < 0.001) with 88.89% (40/45) patients reporting a ≥15 point ODI decrease at 12 months. Mean VAS pain score decrease was 4.31 ± 2.51 at 12 months (*p* < 0.001) and more than 69% reported a 50% reduction in VAS pain scale.	3 non-serious device procedure-related events were reported; one event was an aborted surgery due to an inability to access the pedicle. Two events were for potential pedicle breach with associated radiculitis.
Conger et al. 2022 [24]	Pain Med	Systematic review with single-arm meta-analysis of 12 publications, representing six unique study populations, with 414 participants allocated to receive BVN RFA	Grades of Recommendation, Assessment, Development, and Evaluation (GRADE) framework was used to evaluate the overall quality of evidence. Looked at adults age > 18 with chronic LBP associated with MC1 and MC2 changes on MRI comparing BVN ablation vs. sham, placebo procedure, active standard care treatment, or none	NA	NA	Single-arm meta-analysis showed a success rate of 65% (95% CI 51–78%) and 64% (95% CI 43–82%) for ≥50% pain relief at 6 and 12 months, respectively. Rates of ≥15-point Oswestry Disability Index score improvement were 75% (95% CI 63–86%) and 75% (95% CI 63–85%) at 6 and 12 months, respectively.	NA

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
