# Peer review of "Basivertebral Nerve Ablation for Treatment of Lower Back Pain"

_biomedicines, 2024, doi:10.3390/biomedicines12092046_

Round 1
Reviewer 1 Report
Comments and Suggestions for Authors
Pathological spinal changes can often be multi-causal, hence the need to analyze the effects of various therapies, including those focused on one cause of pain.
The following papers coincide substantially with the article presented for review.
Conger A, Smuck M, Truumees E, Lotz JC, DePalma MJ, McCormick ZL. Vertebrogenic Pain: A Paradigm Shift in Diagnosis and Treatment of Axial Low Back Pain. Pain Med. 2022 Jul 20;23(Suppl 2):S63-S71. doi: 10.1093/pm/pnac081. PMID: 35856329; PMCID: PMC9297155.
Conger A, Burnham TR, Clark T, Teramoto M, McCormick ZL. The Effectiveness of Intraosseous Basivertebral Nerve Radiofrequency Ablation for the Treatment of Vertebrogenic Low Back Pain: An Updated Systematic Review with Single-Arm Meta-analysis. Pain Med. 2022 Jul 20;23(Suppl 2):S50-S62. doi: 10.1093/pm/pnac070. PMID: 35856331; PMCID: PMC9297160.
None of the cited articles mentions whether the analyzed patients had previously undergone physical therapy and kinesitherapy or special methods in rehabilitation. Pharmacological conservative treatment is ineffective because it only works symptomatically. In contrast, well-planned rehabilitation acts causally.
Good results are described in the reviewed article mainly as a reduction in pain. The reported neurological data and quality of life of observed patients are presented only in some studies. Stepping in with a rather invasive technique to interrupt nociception may result in worsening pathological changes in the vertebral bodies and intervertebral discs as well as in the intervertebral joints. This may not be apparent until the results are observed for a longer period of time.
In addition, all of the side effects that occur in patients, which can take on severe forms of dysfunction, are not quantified.
Patients in whom rehabilitation and a change to a more ergonomic lifestyle have been unsuccessful should be referred for ablation procedures. Admittedly, the FDA uses a rather rigorous and described in detail qualification for ablation surgery, nevertheless it is said about six months of conservative treatment, without going into the details of this treatment. It may be that only pharmacological treatment is involved.
It would be good for comparison of data from different studies to create a table, which would include, for example: author (year), study design, time of conservative treatment, time of follow-up after BVN ablation, population characteristics (among others BMI, smoking, osteoporosis), equipment & method, target protocol, results, safety and tolerability, impact and limitations. It would be important to present the nature of residual pain after the ablation procedure, in order to allow for other causes of the condition than only of vertebral origin, and whether surgical monotherapy should be carried out in such cases.
The authors did not state the limitations of the analysis performed. It would be advisable to reduce the amount of repetition found in other papers and take into account in the analysis the reviewer's comments mentioned above.
Author Response
Comments 1: Pathological spinal changes can often be multi-causal, hence the need to analyze the effects of various therapies, including those focused on one cause of pain.
The following papers coincide substantially with the article presented for review.
Conger A, Smuck M, Truumees E, Lotz JC, DePalma MJ, McCormick ZL. Vertebrogenic Pain: A Paradigm Shift in Diagnosis and Treatment of Axial Low Back Pain. Pain Med. 2022 Jul 20;23(Suppl 2):S63-S71. doi: 10.1093/pm/pnac081. PMID: 35856329; PMCID: PMC9297155.
Conger A, Burnham TR, Clark T, Teramoto M, McCormick ZL. The Effectiveness of Intraosseous Basivertebral Nerve Radiofrequency Ablation for the Treatment of Vertebrogenic Low Back Pain: An Updated Systematic Review with Single-Arm Meta-analysis. Pain Med. 2022 Jul 20;23(Suppl 2):S50-S62. doi: 10.1093/pm/pnac070. PMID: 35856331; PMCID: PMC9297160.
None of the cited articles mentions whether the analyzed patients had previously undergone physical therapy and kinesitherapy or special methods in rehabilitation. Pharmacological conservative treatment is ineffective because it only works symptomatically. In contrast, well-planned rehabilitation acts causally.
Good results are described in the reviewed article mainly as a reduction in pain. The reported neurological data and quality of life of observed patients are presented only in some studies. Stepping in with a rather invasive technique to interrupt nociception may result in worsening pathological changes in the vertebral bodies and intervertebral discs as well as in the intervertebral joints. This may not be apparent until the results are observed for a longer period of time.
In addition, all of the side effects that occur in patients, which can take on severe forms of dysfunction, are not quantified.
Patients in whom rehabilitation and a change to a more ergonomic lifestyle have been unsuccessful should be referred for ablation procedures. Admittedly, the FDA uses a rather rigorous and described in detail qualification for ablation surgery, nevertheless it is said about six months of conservative treatment, without going into the details of this treatment. It may be that only pharmacological treatment is involved.
It would be good for comparison of data from different studies to create a table, which would include, for example: author (year), study design, time of conservative treatment, time of follow-up after BVN ablation, population characteristics (among others BMI, smoking, osteoporosis), equipment & method, target protocol, results, safety and tolerability, impact and limitations. It would be important to present the nature of residual pain after the ablation procedure, in order to allow for other causes of the condition than only of vertebral origin, and whether surgical monotherapy should be carried out in such cases.
The authors did not state the limitations of the analysis performed. It would be advisable to reduce the amount of repetition found in other papers and take into account in the analysis the reviewer's comments mentioned above.
Response 1: Thank you so much for giving us your time and valuable feedback. We have worked to improve and revise our manuscript by providing more transparency in the studies and analyses performed along with providing a table that depicts this. We have also tried to incorporate more of the other multidisciplinary factors that can affect patients' pain levels analyzed in said studies.
Reviewer 2 Report
Comments and Suggestions for Authors
The article is indeed interesting and well-written in terms of English language. The authors clearly present an overview of the studies that investigates the efficacy of spinal modic changes and therapy with radiofrequency in patients with chronic low back pain.
However, the article could be improved by providing a clearer explanation of the research objectives, materials, and methods in a dedicated section. This would allow readers to better understand the study design, article selection criteria and data collection procedures. It would be beneficial to provide more information on the inclusion and exclusion criteria for the studies presented. This would enhance the transparency and reliability of the research findings.
Overall, while the article is well-written and informative, I believe that providing more details on the research aim and on the methodology and materials would further enhance its quality and usefulness for readers.
Author Response
Comments 1:
The article is indeed interesting and well-written in terms of English language. The authors clearly present an overview of the studies that investigates the efficacy of spinal modic changes and therapy with radiofrequency in patients with chronic low back pain.
However, the article could be improved by providing a clearer explanation of the research objectives, materials, and methods in a dedicated section. This would allow readers to better understand the study design, article selection criteria and data collection procedures. It would be beneficial to provide more information on the inclusion and exclusion criteria for the studies presented. This would enhance the transparency and reliability of the research findings.
Overall, while the article is well-written and informative, I believe that providing more details on the research aim and on the methodology and materials would further enhance its quality and usefulness for readers.
Response 1: Thank you so much for taking the time to read our draft and provide your useful and valuable feedback. We agree and decided to provide more details on our methodology and research objectives.
Round 2
Reviewer 1 Report
Comments and Suggestions for Authors
The authors have done a great deal of work in revising the text.
It is now presented in accordance with the reviewer's comments.
At the moment the text is suitable for publication.